# Evaluating livestock farmers knowledge, beliefs, and management of arboviral diseases in Kenya: A multivariate fractional probit approach

Paul Nyamweya Nyangau[1,2]*, Jonathan Makau Nzuma[1], Patrick Irungu[1], Menale Kassie[2]

1 Department of Agricultural Economics, Faculty of Agriculture, University of Nairobi, Nairobi, Kenya,
2 International Centre of Insect Physiology and Ecology (icipe), Nairobi, Kenya

* pnyangau@icipe.org

**Data Availability Statement:** The data is held in a public repository: https://datadryad.org/stash/share/ZRuLlzwYZYtCJuEI8Qmbrhj4M2UY5Sb4XPk7dwwSOZA.

## Abstract

Globally, arthropod-borne virus (arbovirus) infections continue to pose substantial threats to public health and economic development, especially in developing countries. In Kenya, although arboviral diseases (ADs) are largely endemic, little is known about the factors influencing livestock farmers' knowledge, beliefs, and management (KBM) of the three major ADs: Rift Valley fever (RVF), dengue fever and chikungunya fever. This study evaluates the drivers of livestock farmers' KBM of ADs from a sample of 629 respondents selected using a three-stage sampling procedure in Kenya's three hotspot counties of Baringo, Kwale, and Kilifi. A multivariate fractional probit model was used to assess the factors influencing the intensity of KBM. Only a quarter of the farmers had any knowledge of ADs while over four-fifths of them could not manage any of the three diseases. Access to information (experience and awareness), income, education, religion, and distance to a health facility considerably influenced the intensity of farmers' KBM of ADs in Kenya. Thus, initiatives geared towards improving access to information through massive awareness campaigns are necessary to mitigate behavioral barriers in ADs management among rural communities in Kenya.

## Author summary

Arboviral infection in humans and animals is on the rise globally due to expansion of vector habitats. Despite the economic and social impact of diseases caused by arboviral infection such as chikungunya, dengue, and Rift Valley fever, little is known in terms of community knowledge, beliefs, and management. Evaluating community knowledge, beliefs, and management practices of arboviral diseases is important for better policy guidance and public health investment. We conducted a survey in Kenya's three hotspot counties of Baringo, Kwale, and Kilifi to understand the factors influencing knowledge, beliefs, and management of arboviral diseases. We found low levels of knowledge and poor managerial skills of arboviral diseases that were largely driven by access to information and

**Funding:** This study has been supported by funding from the Charité -Universitätsmedizin Berlin, Grant number- JU2857/9-1 (received by MK). We also acknowledge the icipe core funding from UK's Foreign, Commonwealth & Development Office (FCDO); the Swedish International Development Cooperation Agency (Sida); the Swiss Agency for Development and Cooperation (SDC); the Federal Democratic Republic of Ethiopia; and the Government of the Republic of Kenya. The funders had no role in study design, data collection and analysis, decision to publish, or preparation of the manuscript.

**Competing interests:** The authors have declared that no competing interests exist.

asset ownership. Thus, community sensitization through improved access to information is important in increasing awareness and increase the management of arboviral diseases among rural communities in Kenya and other sub-Saharan African countries.

## Introduction

Arthropod-borne viruses (arboviruses), known to be transmitted between vertebrate hosts and arthropod vectors, constitute a great concern for global public health [1]. Historically, arboviruses such as chikungunya virus, dengue virus, Rift Valley fever (RVF) virus, yellow fever virus, and zika virus have caused notable diseases leading to animal and human morbidity and mortality [2]. Infections in humans and animals with clinical manifestations could range from subclinical to life-threatening conditions [3]. For example, approximately 96 million symptomatic dengue cases and an estimated 40,000 deaths due to dengue are reported globally every year [4]. The zoonotic effect of arboviral diseases (ADs) include the decline in household income by reducing livestock stock, product sales and consumption, as well as increasing household vulnerability in cases where livestock is used as a risk-coping mechanism [5].

In Kenya, multiple AD outbreaks have resulted in substantial economic losses and public health distress in the past three decades. These include the yellow fever outbreaks of 1992, 1995, and 2016 [2,6,7]; chikungunya fever in 2004 and 2016 [2,8]; RVF incursions in 1997 and 2006 [8,9], and dengue fever outbreaks of 2011–2014 and 2017 [2]. These outbreaks resulted in widespread abortion and death of livestock, and reduced milk production, wool production, livestock growth, working days in humans, and draft animals [10]. In rural communities where agriculture is the dominant livelihood source, the ADs can cause considerable health and economic losses. For example, the 2007 RVF outbreak in Kenya contributed to economic losses estimated at US$32 million [11].

The incidence of ADs is increasing, not just in East Africa but also in many regions of the world. This is due to several factors, including climate change, increased agricultural activity, and ecosystem changes [12]. Global warming, deforestation, and urbanization have led to a rapid expansion of the vectors' habitats and have caused an enormous increase in vector-borne diseases worldwide [3]. Besides, the growing movement across regions of people and livestock has contributed to the broader distribution of the vectors that transmit emerging infectious diseases [13].

The effective management of the ADs depends on people's perceptions of the disease, which in turn, are influenced by the availability of information for decision making as well as the level of knowledge and skills in disease management [14,15]. Previous studies reveal the limited awareness of ADs vectors, signs and symptoms among communities and livestock keepers in East Africa [16–20]. Other studies show poor management regarding ADs [21–23].

Evaluating community knowledge, beliefs, and management practices (KBM) of ADs is relevant for better policy guidance and investment in improving the affected communities' health and economic status. The study on the KBM of ADs is also useful for setting a research agenda and developing targeted communication messages. Although, KBM studies have been undertaken previously in Eastern Africa on RVF [16–21,24], to the best of our knowledge, no study has examined the KBM of a portfolio of ADs (RVF, Chikungunya fever, and Dengue fever) and their drivers in the region.

The failure to examine the KBM in a portfolio format has important implication in terms of accurate risk assessment with impact on the prevention and control of arbovirus infections [25]. Even where KBM studies were undertaken for RVF, the studies used few respondents in

one district. For example, Abdi et al. [21] assessed KBM of RVF among 392 pastoralists living in Ijara district. Similarly, Owange et al. [18] assessed risk factors of RVF among 31 key informants in Ijara district. This study assessed the KBM for three ADs in three hotspot counties in Kenya, namely, Baringo, Kilifi, and Kwale.

Our analysis contributes to the current limited empirical literature on KBM of ADs in the following ways. First, no KBM study has been conducted in the three ADs hotspot counties in the past. Second, our study employs a multivariate probit (MVP) analysis that considers the potential correlation between the KBM across different diseases to assess the socioeconomic and cultural factors that influence household health behavior. Finally, the study used the multivariate fractional probit (MVFP) model that considers the proportion of the correct answers provided by households for each outcome variable to estimate the intensity of KBM. We found low levels of knowledge and poor managerial skills of ADs that were largely driven by access to information and asset ownership.

## Methodology

### Ethics Statement

The protocol for this study was approved by the Scientific and Ethics Review Unit (SERU) of the Kenya Medical Research Institute (KEMRI) reference number 3312. Before the interviews, the study objectives were clearly explained to all research participants and emphasize on voluntary withdraw from the interview at any given time was provided. With the assurance of confidentiality, oral informed consent was obtained from all study participants before the start of the interviews.

### Analytical framework

The Theory of Planned Behaviour (TPB) has been widely used in explaining the relationship between disease management and health-related outcomes [26–28]. The TPB is an extension of the Theory of Reasoned Action (TRA) that explains and predicts human behaviour. The TPB argues that decisions on certain behaviours result from a reasoned process [29]. According to the TPB, three conceptually independent factors determine a person's intention to manage diseases: attitude (A) towards the behaviour of interest (BI); subjective norms (SN); and perceived behavioural control (PBC). These factors can be presented as:

$$\text{BI} = w_1A + w_2SN + w_3PBC \tag{1}$$

Where $w_1$, $w_2$ and $w_3$ are the relative weights of attitudes, subjective norms, and PBC [30].

The TPB posits that a person's attitude ($A$) towards the behaviour of interest is based on readily accessible beliefs regarding the behaviour's likely consequences [31]:

$$\text{A} \propto \sum b_i e_i \tag{2}$$

Where $b$ is the accessible belief for consequence $i$ and $e$ is the subjective evaluation of the outcome.

On the other hand, subjective norms (SN) refer to the perceived social pressure to perform or not to perform the behaviour of interest [30]. Following Ajzen [31], the SN are a function of an individual's normative beliefs ($n$), and the significance ($s$) to comply with the expectations (Eq 3);

$$\text{SN} \propto \sum n_i s_i \tag{3}$$

The PBC is a function of the composite score derived by summing the products of control belief strength ($c$) times perceived power ($p$) over all accessible control factors such as time, skills, money, and other resources expectations [31]:

$$\text{PBC} \propto \sum c_i p_i \tag{4}$$

We assume that the occurrence of one outcome may be conditional on the occurrence of another outcome, with the correlation between them being either positive or negative [32]. In particular, a knowledgeable household might display positive beliefs or sound management practices towards a disease [33,34]. Knowing the disease signs and symptoms can allow timely recognition of the disease when it occurs. Further, households that are knowledgeable about a particular disease may adopt measures to prevent or quickly seek out either human or animal health services when there is an outbreak.

## Empirical model

We employed a MVP model to operationalize Eq 1 and account for the interdependence between the outcome variables [35–38]. Following Young et al. [37], knowledge ($K$), belief ($B$) and management ($M$) of different diseases are a binary function of the decision maker's characteristics and can be modelled using the MVP regression as follows:

$$K = \beta_0^k + \beta_1^k X_1 \ldots \ldots \ldots + \beta_m^k X_m + \epsilon^k, \, K = 1 \text{ if } K > 0, \, 0 \text{ otherwise} \tag{5}$$

$$B = \beta_0^b + \beta_1^b X_1 \ldots \ldots \ldots + \beta_m^b X_m + \epsilon^b, \, B = 1 \text{ if } B > 0, \, 0 \text{ otherwise} \tag{6}$$

$$M = \beta_0^m + \beta_1^m X_1 \ldots \ldots \ldots + \beta_m^m X_m + \epsilon^m, \, M = 1 \text{ if } M > 0, \, 0 \text{ otherwise} \tag{7}$$

where $\beta$ is the vector of parameters to be estimated, $X$ is a vector of decision maker's characteristics [24,34,39], and $\epsilon$ is a vector of the error term.

In the multivariate model, the error terms jointly follow a multivariate normal distribution (MVN) with zero conditional mean and variance, normalized to unity for identification of the parameters, ($\epsilon \sim MNV(0, \Omega)$), where $\Omega$ is the symmetric covariance matrix defined as:

$$\Omega = \begin{bmatrix} 1 & \rho_{BK} & \rho_{MK} \\ \rho_{KB} & 1 & \rho_{MB} \\ \rho_{KM} & \rho_{BM} & 1 \end{bmatrix} \tag{8}$$

where $\rho$ is the unobserved correlation of the KBM equations. A significant $\rho$ indicates interdependence between the error terms. A positive value of $\rho$ is considered "promotive" between the measured pair of equations, while a negative value of $\rho$ is "substitutive" [40]. The STATA command "mvprobit" was used to estimate the parameters $\beta$ and $\rho$.

The MVP model specification measures the determinants of the binary dependent variables (K, B, and M) with no distinction made between respondents that correctly answered one, two, three, or more knowledge-related questions. In other words, it ignores heterogeneity and/ or knowledge intensity differences among the respondents. To correct this anomaly, the MVFP model allows the researcher to assess factors that determine the intensity of KBM. The intensity of each outcome variable is defined as the fraction of the number of correct answers provided by respondents for the sets of questions used in the survey and is estimated by the MVFP by treating those answers as a fractional outcome variable [41]. The MVFP allows the interdependence of the KBM outcome variables.

Because knowledge ($K$), belief ($B$), and management ($M$) are not directly observable, they can be represented by latent variables $K_s^*$, $B_s^*$, and $M_s^*$, that underlie the knowledge, belief, and management status of decision-making units in the sample. Following Schwiebert [42], the relationship between the unobservable latent variable (e.g., $K_s^*$) and the outcome of interest (e.g., $K_s$) can be specified as follows:

$$K_s^* = \beta_0^k + \beta_1^k X_1 \ldots \ldots \ldots \ldots + \beta_n^k X_n + e^k, \; 0 \leq K_s^* \leq 1, \tag{9}$$

$$B_s^* = \beta_0^b + \beta_1^b X_1 \ldots \ldots \ldots \ldots + \beta_n^b X_n + e^b, \; 0 \leq B_s^* \leq 1 \tag{10}$$

$$M_s^* = \beta_0^m + \beta_1^m X_1 \ldots \ldots \ldots + \beta_n^m X_n + e^m, \; 0 \leq M_s^* \leq 1 \tag{11}$$

where $\beta$ and $X_n$ are as previously defined, $K_s$, $B_s$ and $M_s$ are fractional dependent variables that describe the share of total score obtained by the household, and $e^k$, $e^b$, and $e^m$ are disturbance terms assumed to be independent and identical across individual households [41]. The error term, $e = (e^k, e^b, e^m)$ is multivariate normally distributed with a mean vector of zeros and a correlation matrix [42]:

$$\begin{pmatrix} e^k \\ e^b \\ e^m \end{pmatrix} \sim N \left[ \begin{pmatrix} 0 \\ 0 \\ 0 \end{pmatrix} \begin{pmatrix} 1 & \rho_{BK} & \rho_{MK} \\ \rho_{KB} & 1 & \rho_{MB} \\ \rho_{KM} & \rho_{BM} & 1 \end{pmatrix} \right] \tag{12}$$

In this study, the unknown parameters $\beta$ and $\rho$ were estimated using a seemingly unrelated regression with ordered responses [43], under the conditional mixed process estimator with multilevel random effects command "cmp" available in STATA software.

## Study area and sampling procedure

This study was carried out in the three ADs hotspot counties of Baringo, Kilifi, and Kwale in Kenya (Fig 1). Baringo is prone to floods leading to outbreaks of arboviral diseases. For instance, the 1997/98 El-Niño rains resulted in an episode of yellow fever, while the 2006/07 heavy rains resulted in an outbreak of RVF [44]. Kwale and Kilifi are areas from where the chikungunya virus started before spreading to other parts of the country, representing one of the critical seeding regions for ADs. Malaria, dengue fever, chikungunya fever, and lymphatic filariasis are common mosquito-borne diseases in the two areas [45]. Initially, focus group discussions were conducted in the study sites to determine the most important ADs and to adjust the survey tool. According to community members living in the three study sites, RVF, Chikungunya fever, and Dengue fever were the most prevalent ADs. Later, a multistage sampling technique was used to select 629 respondents for a survey of their KBM of ADs in their locale. In the first stage, the three ADs hotspot counties (Baringo, Kilifi, and Kwale) were purposively selected. In the second stage, purposive sampling was also used to select the most ADs-prone subcounties (decentralized units within a county) in each of the three counties resulting in three study sites of Marigat in Baringo, Malindi in Kilifi and Msambweni in Kwale. A sampling frame of all households in the three study sites was obtained from the local administration (chiefs and village elders). In the third stage, a simple random sampling technique was used to select 200 households from each study site giving a total sample of 629 households after adjusting for 10 percent of the non-responses following Mutiso [46]. Well-trained enumerators undertook face-to-face interviews through a pre-tested semi-structured questionnaire using CSPro version 7.5 electronic data collection software [47].

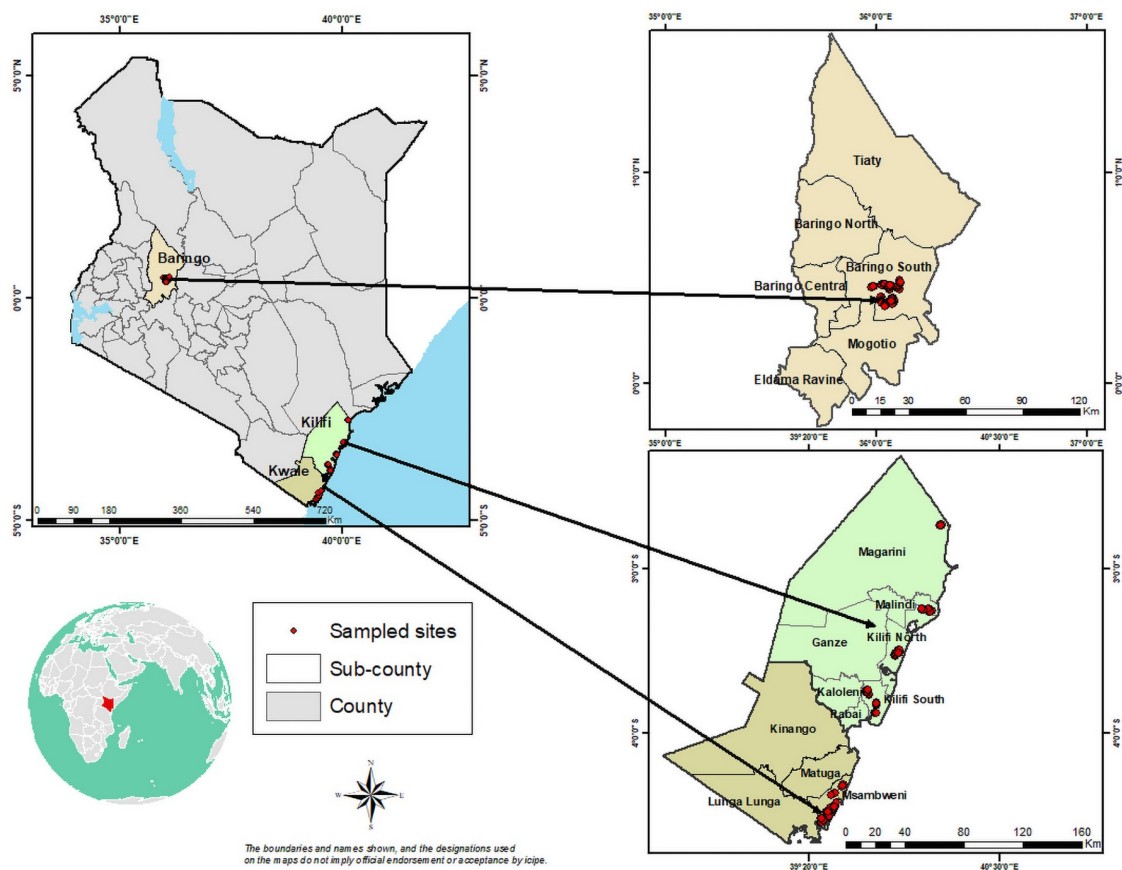

**Fig 1. Study area and sampled households: Emily Kimathi, GIS unit, icipe; The map was developed using the QGIS 3.16 software, https://qgis.org/en/site/forusers/download.html.**

### Measurement of variables

**Outcome variables.** The outcome variables of interest in this study included knowledge, beliefs, and management practices of ADs. These variables were measured as dummy variables. The knowledge score was constructed using 55 binary response questions for RVF and 14 each for chikungunya and dengue fevers. A total of 8 and 7 binary response questions were used to generate the beliefs and management scores for the three ADs. The beliefs section consisted of the perceived threat associated with ADs. The management practices were related to a group of actions taken to prevent the spread of ADs [22]. A respondent was considered knowledgeable, to have positive beliefs, and as having good ADs management practices of the three ADs when they correctly answered 50 percent of the questions posed under each outcome (*KBM*) variable. Based on this, the outcome variables took a value of one if the respondent answered 50 percent of the questions correctly and zero otherwise (i.e., having either an incorrect response, answering "I don't know", or having missing answers). The fractional variable used in the MVFP model was constructed as the sum of correct answers to knowledge, belief, and management practices questions as a ratio of the total number of questions asked per outcome variable. For instance, the intensity of knowledge of RVF was measured as the number of correct answers as a share of the 55 knowledge questions of RVF. The intensity of knowledge of dengue and chikungunya fever were measured as the number of correct answers as a share of 14 questions, Moreover, the intensity of beliefs on each of the three ADs was measured

as a ratio of the correct answers to a total 8 questions. Finally, the intensity of management of the three ADs was measured as the number of correct answers as a share of 7 questions.

**Explanatory variables.** The choice of the explanatory variables used in this study was informed by previous studies [21,24,33,34,39]. These variables included access to health information, social capital and networks, asset endowment and household demographic characteristics. Three variables were used to measure "access to health information", namely, distance to the nearest health facility, awareness of health impacts of ADs, and household experience with an AD. The distance between the homestead and the nearest health facility measured in the amount of time it takes to walk between the two points was used as proxy for access to health information. The variable awareness of health impacts took a value of one if the respondent understood the health impacts of ADs, and zero otherwise.

Experience with an AD was measured as a dummy variable with a value of one if a family member had suffered from any AD in the 12 months preceding the survey and zero otherwise. Social capital and networking was proxied by group membership measured as a dummy variable with the value of 1 if the respondent was a member of a health promotion group and 0 otherwise. Health promotion group constituted individuals who receive training from health specialists, thus increasing awareness of good health, diet, and exercise in the society. The number of tropical livestock units (TLU) kept by a household was used as a proxy for asset ownership in this study. The heterogeneity of the households was controlled in the regression model by including the household head's education level, gender, and religion. The level of formal education attained was measured as the number of years of formal schooling completed by the household head. The gender of the household head was measured as a dummy variable, taking a value of one if the household head was male and zero otherwise. In this study, religion was measured as a categorical variable coded 0, 1 and 2 for other religions, Christianity, and Islam, respectively.

## Results

### Descriptive Results

The socioeconomic characteristics of the 629 households in the three ADs hotspots in Kenya show that 55 percent of the households were aware of the health impacts of ADs while 33 percent of the households had suffered from at least one of them (Table 1). Christianity was the dominant religion, as reported by 63 percent of the respondents, even though almost all respondents (98 percent) in Kwale were Muslims. On the average, Baringo and Kilifi residents took longer (37 and 35 minutes respectively) to reach the nearest health facility as compared to Kwale residents (24 minutes). Eleven percent of Kwale households belonged to a health promotion group as compared to five percent in the other two study sites. The average number of livestock owned in the study area was 3.69.

The summary statistics of the 629 farmers KBM of ADs in Kenya showed that 16, 29, and 18 percent of respondents had good knowledge of RVF, chikungunya fever, and dengue fever infections, respectively (Table 2). The highest knowledge of RVF was recorded in Baringo (20 percent), while Kwale (36 percent) and Kilifi (23 percent) residents had the highest knowledge of chikungunya fever and dengue fever, respectively. Despite the low knowledge, most respondents had positive beliefs about the three ADs. In Kwale County, 75 and 55 percent of respondents believed that chikungunya and dengue fever, respectively, were dangerous diseases as compared to 44 and 38 percent of the respondents in Kilifi County. The low level of knowledge in the study areas translated into poor management of ADs that ranges between 6 percent of RVF and 27 percent of chikungunya fever.

**Table 1. Characteristics of livestock keepers in Kenya's ADs Hotspots.**

| *Variables* | | Baringo | Kwale | Kilifi | Overall |
|---|---|---|---|---|---|
| | | n = 211 | n = 218 | n = 200 | n = 629 |
| *Household demographic characteristics* | | | | | |
| Education | Household head's years of formal education | 7.06[a] | 6.50[a] | 7.09[a] | 6.88 |
| | | (4.52) | (4.02) | (4.04) | (4.20) |
| Gender | Sex of the household head (1 = male 0 = female) | 0.85[a] | 0.78[a] | 0.83[a] | 0.82 |
| Religion | Religion of respondent (1 = Yes, 0 = No) | | | | |
| Others | | 0[a] | 0[a] | 0.05[b] | 0.02 |
| Christianity | | 0.99[c] | 0.02[a] | 0.92[b] | 0.63 |
| Islam | | 0[a] | 0.97[c] | 0.04[b] | 0.35 |
| *Access to health information* | | | | | |
| Experience | Household suffered from RVF (1 = Yes, 0 = No) | 0.43[b] | 0.00[a] | 0.06[a] | 0.30 |
| | Household suffered from chikungunya fever (1 = Yes, 0 = No) | 0.00[a] | 0.50[a] | 0.43[a] | 0.47 |
| | Household suffered from dengue fever (1 = Yes, 0 = No) | - | 0.21[a] | 0.25[a] | 0.23 |
| Awareness | Household aware of RVF health impacts (1 = Yes, 0 = No) | 0.87[c] | 0.00[a] | 0.25[b] | 0.63 |
| | Household aware of health impacts of chikungunya fever (1 = Yes, 0 = No) | 0.00[ab] | 0.71[b] | 0.50[a] | 0.62 |
| | Household aware of health impacts of dengue fever (1 = Yes, 0 = No) | - | 0.45[a] | 0.31[a] | 0.40 |
| Distance | Distance to the nearest health facility (Walking minutes) | 36.50[b] | 24.32[a] | 34.61[b] | 31.67 |
| | | (35.91) | (25.84) | 25.84 | 30.05 |
| *Social capital and networking* | | | | | |
| Group membership | Whether a member of the household belongs to a health promotion group (1 = Yes 0 = No) | 0.05[a] | 0.11[b] | 0.05[a] | 0.07 |
| *Asset endowment* | | | | | |
| Livestock | Livestock ownership in Tropical livestock unit (TLU) | 8.36[b] | 1.36[a] | 1.29[a] | 3.69 |
| | | (13.28) | (6.16) | (1.87) | (9.18) |
| Income | Total income from all enterprises (KES/Year) 000 | 112.62[a] | 169.50[a] | 145.23[a] | 142.70 |
| | | (142.93) | (443.67) | (264.76) | (312.44) |

*Notes*: Standard deviation in parenthesis; 1US$ = KES 102 at the survey time; Means in the same row, followed by the same letters, are not significantly different at 5%.

**Table 2. Farmers knowledge, beliefs and management of ADs in Kenya.**

| Variable | Description | Baringo | Kwale | Kilifi | Overall |
|---|---|---|---|---|---|
| **Outcome variables** | *Dummy (1 = yes if half of the number of questions were correctly answered)* | | | | |
| **RVF** | | n = 207 | n = 23 | n = 89 | n = 319 |
| Knowledge | Knowledgeable of RVF | 0.20[b] | 0.00[a] | 0.09[a] | 0.16 |
| Beliefs | Positive beliefs towards RVF | 0.88[c] | 0.00[a] | 0.33[b] | 0.66 |
| Management | Have good management practices to prevent RVF | 0.04[a] | 0.09[a] | 0.09[a] | 0.06 |
| **Chikungunya fever** | | n = 1 | n = 191 | n = 145 | n = 337 |
| Knowledge | Knowledgeable of Chikungunya fever | 0.00[ab] | 0.36[b] | 0.19[a] | 0.29 |
| Beliefs | Positive beliefs towards Chikungunya fever | 0.00[ab] | 0.75[b] | 0.55[a] | 0.66 |
| Management | Have good management practices to prevent Chikungunya fever | 0.00[ab] | 0.35[b] | 0.17[a] | 0.27 |
| **Dengue fever** | | n = 0 | n = 84 | n = 52 | n = 136 |
| Knowledge | Knowledgeable of Dengue fever | - | 0.15[a] | 0.23[a] | 0.18 |
| Beliefs | Positive beliefs towards Dengue fever | - | 0.44[a] | 0.38[a] | 0.42 |
| Management | Have good management practices to prevent Dengue fever | - | 0.23[a] | 0.17[a] | 0.21 |

*Notes*: Standard deviation in parenthesis; Means in the same row, followed by the same letters, are not significantly different at 5%.

The study of farmer's specific knowledge of signs, symptoms, transmission methods, beliefs and management practices of the three ADs in Kenya showed that the main signs of the three ADs in human were fever, abdominal pains and headache as reported by 82, 29 and 19 percent of the respondents, respectively (Table 3). In animals, the major signs of RVF were bloody diarrhea, bloody discharge and death among young ones as reported by 58, 57 and 56 percent of the respondents. Over 80 percent of respondents correctly identified mosquitoes as the vectors of the three diseases. Direct contact with blood and other body tissues of infected animals/humans were reported as common methods of transmission of RVF and dengue fever by 48 and 13 percent of the respondents, respectively. When asked about the *Aedes* mosquito's breeding grounds, less than 40 percent of the respondents indicated that mosquitoes breed in water containers. Most respondents (70 percent) did not know when the *Aedes* mosquito bites and incorrectly identified nighttime as the biting time. In comparison, less than two percent of the respondents correctly indicated that chikungunya and dengue vectors bite during the day.

Mosquito nets were the most widely used method of preventing mosquito bites as reported by over 90 percent of the respondents. Over half of the respondents reported bush clearing of overgrown vegetation across the homestead as the most prevalent method used in the control of mosquito breeding whereas covering water-holding containers and/or their proper disposal was reported by only a quarter of the respondents. Eighty and 64 percent of respondents respectively, reported that the management of RVF was the responsibility of the Veterinary Department and the Ministry of Health. Treatment in hospitals was the most dominant management practice followed by purchasing drugs in pharmaceutical outlets, using traditional treatment, and using local herbs.

## Econometric results

The correlation coefficients of the error terms of the multivariate probit MLE estimates of the drivers of KBM of ADs in Kenya were analyzed (Table 4). The likelihood ratio rejects the null hypothesis of no correlation between the three equations' error terms. This confirms the use of the MVP model instead of binary choice models. Some of the pair-wise correlation coefficients between the error terms in the KBM equations were significant, which further supports the MVP model. Knowledge complements beliefs in all three diseases. We estimated MVP and MVFP specifications and find the statistically significant variables in both models to be the same. For brevity, we only presented the results of the MVFP model.

The Multivariate Fractional Probit (MVFP) maximum likelihood estimates (MLE) of the intensity of livestock farmers KBM of ADs in Kenya was estimated (Table 5). Access to information (experience and awareness), income, and some household characteristics (education, and religion) positively and significantly influence the intensity of livestock farmers KBMs on ADs at least at the 5 percent level (P<0.05). In conformity with the expectations, awareness of the health impacts of ADs positively influenced the intensity of livestock farmers KBM of all three ADs in Kenya and was significant at least at the 5 percent level except for the management of RVF. Income had positive significant influences on the intensity of livestock farmers KMB of the three ADs in Kenya at least at the 5 percent level except for the case of the beliefs on RVF.

Education positively influenced the intensity of livestock farmer's knowledge and management practices of the three ADs with the exception of dengue fever and was significant at least at the 5 percent level. Belonging to the Muslim faith positively influenced the intensity of livestock farmers' beliefs on RVF and was significant at the 1 percent level. While being a Christian reduced the intensity of livestock farmer's beliefs regarding dengue fever, being a Muslim increased the intensity of livestock farmers' beliefs towards RVF. However, belonging to a

**Table 3. Farmers knowledge of signs, symptoms, transmission methods, beliefs and management of ADs in Kenya.**

| Characteristics | RVF | | Chikungunya fever | | Dengue fever | | Overall |
|---|---|---|---|---|---|---|---|
| | Number of observations (N) | % | N | % | N | % | % |
| Have you heard about this disease? | 629 | 51 | 629 | 54 | 629 | 22 | 42 |
| Main signs and symptoms in humans | | | | | | | |
| Fever | 250 | 72 | 273 | 85 | 71 | 89 | 82 |
| Generalized weakness | 250 | 43 | | | | | 43 |
| Bleeding from nose and gums | 250 | 11 | | | | | 11 |
| Skin rashes | | | | | 71 | 10 | 10 |
| Back pain | 250 | 25 | | | | | 25 |
| Nausea/vomiting | 250 | 8 | | | 71 | 44 | 26 |
| Joint pain | | | 273 | 32 | | | 32 |
| Abdominal pain | 207 | 28 | 272 | 25 | 71 | 34 | 29 |
| Pain behind the eyes | 250 | 2 | 272 | 12 | 71 | 8 | 7 |
| Abortion in pregnant women | 250 | 10 | | | | | 10 |
| Inflammation of brain-headaches, coma, seizures | 250 | 20 | 272 | 13 | 71 | 24 | 19 |
| Fatigue | | | 272 | 34 | | | 34 |
| Main signs and symptoms in animals | | | | | | | |
| Abortion | 249 | 33 | | | | | 33 |
| Bloody Discharge | 249 | 57 | | | | | 57 |
| High fever | 249 | 54 | | | | | 54 |
| Bloody Diarrhea | 249 | 58 | | | | | 58 |
| Death Among young animals | 249 | 56 | | | | | 56 |
| Is mosquito responsible for ADs transmission? | 165 | 82 | 200 | 96 | 54 | 94 | 91 |
| Direct contact with blood and other body tissues from an infected person/animal | 249 | 48 | | | 71 | 13 | 31 |
| Do mosquitoes breed in water containers | 319 | 29 | 337 | 35 | 136 | 29 | 31 |
| When are the *Aedes* mosquitoes most likely to feed/bite? | | | | | | | |
| Nighttime | 319 | 59 | 337 | 77 | 136 | 75 | 70 |
| Day time | 319 | 1 | 337 | 0 | 136 | 1 | 1 |
| Both day and night | 319 | 39 | 337 | 23 | 136 | 24 | |
| Awareness of the methods to prevent mosquito breeding | | | | | | | |
| Clearing bushes around the house | 319 | 58 | 336 | 62 | 136 | 60 | 60 |
| Creating proper drainage of water around the home | 319 | 31 | 336 | 40 | 136 | 46 | 39 |
| Covering water holding containers tightly | 319 | 20 | 336 | 28 | 136 | 32 | 27 |
| Proper disposal of discarded containers | 319 | 16 | 336 | 25 | 136 | 23 | 21 |
| Methods used to control mosquito bites | | | | | | | |
| Mosquito bed nets | 319 | 96 | 336 | 98 | 136 | 99 | 98 |
| Mosquito repellants | 319 | 38 | 336 | 53 | 136 | 58 | 50 |
| Indoor residual insecticides spraying | 319 | 35 | 336 | 52 | 136 | 65 | 51 |
| Screening/fencing windows and doors | 319 | 15 | 336 | 37 | 136 | 43 | 32 |
| Close doors and windows by 6.00PM | 319 | 44 | 336 | 38 | 136 | 35 | 39 |
| Plants to repel Mosquitoes | 319 | 6 | 336 | 10 | 136 | 8 | 8 |
| Player of a major role in the control of this disease | | | | | | | |
| Veterinary Authority | 249 | 80 | 272 | 4 | 71 | 1 | 28 |
| Health Authority | 249 | 64 | 272 | 90 | 71 | 97 | 84 |
| Environmental Authority | 249 | 7 | 272 | 16 | 71 | 17 | 13 |
| Community | 249 | 29 | 272 | 24 | 71 | 32 | 28 |

*(Continued)*

**Table 3.** (Continued)

| Characteristics | RVF | | Chikungunya fever | | Dengue fever | | Overall |
|---|---|---|---|---|---|---|---|
| | Number of observations (N) | % | N | % | N | % | % |
| What would your household do if you suspect that you or your family member has been infected with this disease? | | | | | | | |
| Local herb (e.g. pawpaw leaves to treat Chikungunya) | 250 | 6 | 273 | 24 | 71 | 1 | 10 |
| Chemist medicine | 250 | 5 | 273 | 7 | 71 | 6 | 6 |
| Seek traditional treatment | 250 | 8 | 273 | 8 | 71 | 4 | 7 |
| Seek treatment in hospital | 250 | 85 | 273 | 88 | 71 | 92 | 88 |

health group and the distance to the nearest health facility negatively influenced the intensity of livestock farmers' beliefs and management practices of dengue and RVF, respectively, and were both significant at the 5 percent level.

## Discussion

Slightly above half of the households in the study were aware of the three ADs while a third of them had experienced at least one of the ADs (a family member had suffered from at least one of the ADs). Most households in Baringo were aware of RVF but none of them was aware of the health impacts of chikungunya and dengue fever. On the other hand, majority of the households in Kwale and Kilifi Counties were aware of the health impacts of chikungunya and dengue fever. More residents in Baringo County experienced RVF infections among family members as compared to their counterparts in Kilifi and Kwale counties. Similarly, more

**Table 4.  Correlation coefficients of the error terms of the MVP estimates.**

| Disease | $\rho$K | $\rho$B | $\rho$M |
|---|---|---|---|
| RVF | | | |
| $\rho$K | 1 | | |
| $\rho$B | 0.640 (0.136)** | 1 | |
| $\rho$M | -0.371 (0.216) | -0.171 (0.190) | 1 |
| Likelihood ratio test of $\rho$KB = $\rho$KM = $\rho$BM = 0: $\chi^2$ (3) = 8.500, Prob > $\chi^2$ = 0.037 | | | |
| Chikungunya fever | | | |
| $\rho$K | 1 | | |
| $\rho$B | 0.364 (0.114)** | 1 | |
| $\rho$M | 0.538 (0.084)*** | 0.233 (0.111)** | 1 |
| Likelihood ratio test of $\rho$KB = $\rho$KM = $\rho$BM = 0: $\chi^2$ (3) = 42.558, Prob > $\chi^2$ = 0.000 | | | |
| Dengue fever | | | |
| $\rho$K | 1 | | |
| $\rho$B | 0.650 (0.165)** | 1 | |
| $\rho$M | 0.253(0.173) | 0.216 (0.173) | 1 |
| Likelihood ratio test of $\rho$KB = $\rho$KM = $\rho$BM = 0: $\chi^2$ (3) = 7.002, Prob > $\chi^2$ = 0.072 | | | |

Notes: Standard errors in parenthesis; K = Knowledge, B = Beliefs, M = Management;

* = significant at p < 0.1;

** = significant at p < 0.05;

*** = significant at p < 0.00

**Table 5. Multivariate fractional probit maximum likelihood estimates of the intensity of farmers KBM of ADs in Kenya.**

| Variables | RVF | | | Chikungunya fever | | | Dengue fever | | |
|---|---|---|---|---|---|---|---|---|---|
| | Knowledge | Beliefs | Management | Knowledge | Beliefs | Management | Knowledge | Beliefs | Management |
| *Household demographic characteristics* | | | | | | | | | |
| Education | 0.010* | 0.006 | 0.024*** | 0.006 | -0.015 | 0.024*** | 0.004 | -0.005 | 0.031** |
| | (0.006) | (0.009) | (0.006) | (0.005) | (0.010) | (0.007) | (0.010) | (0.023) | (0.012) |
| Gender | -0.097* | -0.045 | -0.112 | -0.054 | -0.030 | -0.080 | -0.007 | -0.190 | -0.077 |
| | (0.054) | (0.115) | (0.070) | (0.043) | (0.091) | (0.060) | (0.084) | (0.204) | (0.090) |
| Christianity | -0.165 | 0.325 | -0.207 | -0.086 | -0.060 | 0.054 | -0.045 | -0.768* | -0.015 |
| | (0.209) | (0.309) | (0.170) | (0.184) | (0.207) | (0.122) | (0.378) | (0.417) | (0.198) |
| Islam | 0.138 | 5.156*** | -0.130 | -0.112 | -0.390 | -0.066 | -0.619 | -0.279 | -0.089 |
| | (0.215) | (0.370) | (0.181) | (0.217) | (0.265) | (0.138) | (0.379) | (0.442) | (0.232) |
| *Access to health information* | | | | | | | | | |
| Experience | 0.230*** | 0.240** | 0.079 | 0.080** | 0.547*** | -0.069 | 0.092 | 0.513** | 0.036 |
| | (0.044) | (0.082) | (0.066) | (0.038) | (0.077) | (0.052) | (0.099) | (0.161) | (0.099) |
| Awareness | 0.314*** | 0.762*** | 0.110 | 0.365*** | 0.739*** | 0.339** | 0.482*** | 1.544*** | 0.414*** |
| | (0.073) | (0.103) | (0.098) | (0.041) | (0.090) | (0.058) | (0.080) | (0.180) | (0.087) |
| Distance | 0.012 | -0.050 | -0.061** | -0.015 | -0.018 | -0.034 | 0.066* | 0.110 | 0.021 |
| | (0.021) | (0.035) | (0.031) | (0.019) | (0.043) | (0.028) | (0.036) | (0.083) | (0.045) |
| *Social capital and networks* | | | | | | | | | |
| Group membership | -0.003 | 0.060 | 0.078 | -0.011 | 0.150 | -0.051 | -0.083 | -0.472** | -0.011 |
| | (0.102) | (0.162) | (0.126) | (0.066) | (0.154) | (0.100) | (0.080) | (0.208) | (0.112) |
| *Asset endowment* | | | | | | | | | |
| Livestock units | 0.022 | -0.005 | -0.011 | | | | | | |
| | (0.018) | (0.030) | (0.022) | | | | | | |
| Income | 0.046** | 0.058 | 0.126*** | 0.048*** | 0.123*** | 0.041** | 0.067** | 0.213** | 0.084** |
| | (0.020) | (0.038) | (0.026) | (0.012) | (0.034) | (0.017) | (0.029) | (0.071) | (0.034) |
| *Location fixed effects* | | | | | | | | | |
| Kwale | -0.988*** | -6.859*** | 0.000 | -0.284** | 6.324*** | 0.199 | | | |
| | (0.126) | (0.434) | (0.229) | (0.140) | (0.527) | (0.155) | | | |
| Kilifi | -0.276** | -0.631*** | 0.048 | -0.454*** | 5.840*** | -0.122 | -0.153 | 0.404** | -0.154 |
| | (0.083) | (0.138) | (0.111) | (0.090) | (0.504) | (0.124) | (0.102) | (0.176) | (0.147) |
| Constant | -1.050*** | -1.070** | -2.036*** | -0.613** | | -0.929** | -1.063** | -3.624*** | -1.626*** |
| | (0.299) | (0.512) | (0.340) | (0.256) | | (0.279) | (0.506) | (0.989) | (0.451) |
| Wald statistics | $\chi^2$ (36) = 1834.52, Prob > $\chi^2$ = 0.000 | | | $\chi^2$ (33) = 62028.06, Prob > $\chi^2$ = 0.000 | | | $\chi^2$ (30) = 294.08, Prob > $\chi^2$ = 0.000 | | |
| Observations | 276 | | | 334 | | | 136 | | |

Notes: Other religion used as the base in the religion category; Baringo used as the base in the location fixed effects category (RVF and chikungunya) while Kwale is used as the base in the case of dengue fever in the location fixed effects category; Confidence intervals (95%) in parenthesis;

* = significant at p < 0.1;

** = significant at p < 0.05;

*** = significant at p < 0.00.

households in Kwale and Kilifi counties had family members who had suffered from chikungunya and dengue fever as compared to households in Baringo County. These findings support Atoni et al. [2] who have reported RVF to be endemic in Baringo County while Kwale and Kilifi counties are hotspots of both chikungunya and dengue fever.

With regards to the KBM, we found that most respondents had poor knowledge of the three ADs. However, many households believed that the three ADs were dangerous diseases.

This could be explained by the fact that most people in the society perceive diseases as a real threat despite having little knowledge about them. Good management skills of the three ADs were reported only by a few of the respondents in Kenya. Fever was the most prevalent sign and symptom of ADs among humans. This was consistent with earlier studies that have reported fever as the most frequently stated symptom of RVF [21], chikungunya fever [48], and dengue fever [33]. Other disease signs and symptoms mentioned by a few respondents included abdominal pains and headaches among humans and bloody diarrhea, bloody discharge, and death of young animals. These findings suggest that most of the respondents had limited knowledge of these AD signs, symptoms, and methods used to control their spread.

The most widely used methods of mosquito control reported by respondents in declining order of importance included use of nets, bush clearing, covering water-holding containers and/or their proper disposal. However, mosquito nets may offer little protection in reducing the risk of ADs. This is because most ADs mosquitoes feed during the day when households are not using the nets [49,50]. Similar findings have been reported in Kenya [21] and Pakistan [39]. Lack of such knowledge, especially in areas with a high density of *Aedes aegypti*, poses a challenge in ADs prevention. Almost all respondents reported that management of ADs was the responsibility of the Veterinary Department and the Ministry of Health. Expecting that government departments would control ADs might hinder community-based efforts towards controlling their spread leading to increased risk of infection. Bartumeus et al. [51] highlighted the importance of local communities in vector control. Most respondents indicated they will visit hospital when they suspect of ADs infection. These findings are consistent with other studies such as Kumaran et al. [52] and Nguyen et al. [23] that have reported hospital health-seeking behaviour in managing ADs.

We found positive significant influences of access to information (experience and awareness), income, education, distance to a health facility and religion on the intensity of livestock farmers' KBM of ADs in Kenya. In conformity with the expectations, information access (experience and awareness) increased livestock farmer's intensities of KBM of ADs in Kenya. Awareness of the health impact of the three ADs increased the intensity of livestock farmers KBM by between 31 and 154 percent (Table 5). Individuals who were aware of AD health impacts were more likely to undertake disease mitigation strategies or seek medical intervention [53]. Overall, farmers who were aware of the attributes of any intervention were more likely to have favourable management practices than their counterparts who were not aware [54].

The study identified that a household head's education level increased the intensity of knowledge, beliefs, and management of ADs, which was consistent with previous studies [22,23,52,55]. An extra year spent in school increased a livestock farmer's intensity of management practices of RVF and chikungunya fever by two percent while that of dengue fever increased by three percent. Education improves access to information and provides individuals with the ability to interpret and implement different disease management strategies [34]. Education provides good knowledge of disease signs and symptoms as illustrated by Khun and Manderson [56] which is important for timely disease prevention. The relationship between education and management of ADs has been documented in other studies [23,33,34,57].

Experience of the health impacts of a disease is important in influencing the management of ADs. Households that had experienced RVF and chikungunya fever had a higher intensity of knowledge and beliefs of both diseases by between 8 and 54 percent as compared to their counterparts who did not have at least a family member who had suffered from any of the three ADs (Table 5). This finding was consistent with the findings of Abdi et al. [21] and Harapan et al. [34] that demonstrated a positive and significant relationship between household's experience and knowledge and beliefs of ADs. Income was associated with increased livestock farmer intensities of KBM on ADs in Kenya. A one percent increase in household income

increased livestock farmers intensities of KBM of the three ADs by between 4 and 13 percent. The possible reason for a positive association between the intensity of KBM and income is that people with higher economic status might have better information access on ADs and resources to manage the diseases [58,59]. A direct relationship between income and good knowledge of dengue has been documented [59–61]. Similarly, Alhoot et al. [62], Ghani et al. [63], and Lugova and Wallis [61] have reported a significant association between income and positive beliefs regarding dengue fever.

Farmers who were located further away from health facilities had poor management skills of ADs as compared to their counterparts who had better access to medical facilities. An extra minute spent walking to the health facility to seek treatment reduced the intensity of a livestock farmer's management skills of RVF by six percent. This suggested that as distance increased, the likelihood of the household members visiting health facilities declined and thus they were less likely to manage the diseases. Health facilities are the principal point for sourcing health information in many rural settings through the distribution of education materials on signs and symptoms and prevention methods of a diseases [64]. Feikin et al. [65] has documented a negative relationship between distance of residence from a health facility and utilisation of health services in Kenya.

Being a Muslim increased the intensity of a livestock farmers' beliefs that RVF is a dangerous disease by a huge margin of 516 percent (Table 5). Religion positively or negatively influences people's beliefs regarding their willingness to receiving a certain treatment [66]. Similar finding has been documented by Chandren et al. [67] and Harapan et al. [34] which indicated that religion influenced people's beliefs regarding dengue fever in Malaysia and Indonesia.

Though our study generates important information, the study has the following caveats. First, our results must be interpreted with caution since the relationships are based on one point and do not account for the relationship dynamics of the factors analyzed. Therefore, we cannot construe the relationships between knowledge, beliefs, management, and associated factors. Secondly, our study conducted interviews using a semi-structured questionnaire; thus, some questions, especially on beliefs, might have been influenced by the respondent's social desires. Finally, despite its wide-spread use, the TBP does not account for other factors that might influence intention and motivation of individuals [68]. Nevertheless, this study provides an insight into the knowledge, beliefs, and management of people regarding RVF, chikungunya fever, and dengue fever in Kenya.

## Conclusions and policy implications

This study evaluated the intensity of livestock farmer's knowledge, beliefs, and management of RVF, chikungunya, and dengue fever in Kenya using a MVFP model that employs a sample of 629 households. Slightly above half of the respondents were aware of the three ADs, while a third of the respondents had experienced (suffered) the ADs. While only a small share of the respondents have basic knowledge about the three diseases, a vast majority of them considered ADs as serious diseases affecting both animals and humans. Despite the low knowledge, more than half of the respondent expressed positive beliefs towards ADs. There was a low translation of knowledge about disease transmission and prevention into good management practices.

We demonstrated the importance of access to information (experience, awareness, and distance to a health facility), income, education, and religion in influencing the KBM of ADs. These findings demonstrate the importance of access to information in influencing the intensity of livestock farmers KBM of ADs. Thus, policy initiatives should focus on increasing livestock farmers' awareness of the three ADs in Kenya to mitigate their negative health impacts. Moreover, the awareness programs on these three ADs should also target different religions

separately. Most importantly, ADs prevention and control should be promoted among individuals who have experienced the diseases, their families, and visiting neighbors by hospitals to raise awareness among community members and use them as outreach program. This will increase the knowledge of ADs' and improve the management of these diseases in the society.

## Acknowledgments

We gratefully acknowledge Dr. David Tchouassi from *icipe* for his insightful comments during the manuscript preparation

## Author Contributions

**Conceptualization:** Menale Kassie.

**Data curation:** Paul Nyamweya Nyangau.

**Formal analysis:** Paul Nyamweya Nyangau, Jonathan Makau Nzuma, Patrick Irungu, Menale Kassie.

**Funding acquisition:** Menale Kassie.

**Investigation:** Paul Nyamweya Nyangau, Jonathan Makau Nzuma, Patrick Irungu, Menale Kassie.

**Methodology:** Paul Nyamweya Nyangau, Jonathan Makau Nzuma, Patrick Irungu, Menale Kassie.

**Project administration:** Menale Kassie.

**Resources:** Menale Kassie.

**Software:** Paul Nyamweya Nyangau, Jonathan Makau Nzuma, Patrick Irungu, Menale Kassie.

**Supervision:** Jonathan Makau Nzuma, Patrick Irungu, Menale Kassie.

**Validation:** Paul Nyamweya Nyangau.

**Visualization:** Paul Nyamweya Nyangau.

**Writing – original draft:** Paul Nyamweya Nyangau.

**Writing – review & editing:** Paul Nyamweya Nyangau, Jonathan Makau Nzuma, Patrick Irungu, Menale Kassie.

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
