## [Decision Letter · Decision Letter 0]

5 Jul 2021

Dear Nyangau,

Thank you very much for submitting your manuscript "Evaluating Knowledge, Beliefs, and Management of Arboviral Diseases in Kenya: A Multivariate Fractional Probit Approach" for consideration at PLOS Neglected Tropical Diseases. As with all papers reviewed by the journal, your manuscript was reviewed by members of the editorial board and by several independent reviewers. In light of the reviews (below this email), we would like to invite the resubmission of a significantly-revised version that takes into account the reviewers' comments. 

We cannot make any decision about publication until we have seen the revised manuscript and your response to the reviewers' comments. Your revised manuscript is also likely to be sent to reviewers for further evaluation.

Sincerely,

Alberto Novaes Ramos Jr

Associate Editor

Victor S. Santos

Deputy Editor

Reviewer's Responses to Questions

**Summary and General Comments**

Reviewer #1: The authors did a very good job with this study at a time when the required information was lacking. This study will serve as guide to future similar studies due to the detailed explanations of the steps and equations used.

Reviewer #2: Overall, this is a very well written manuscript with a very precise explanation of the methods used and the results found. Further, the discussion has put the the studies finding in the context of the current literature and has discussed public health relevance. 

Some mentioning of ethical procedure should be added, such as ethical study approval of the institution.

Further, minor changes are attached in the attached document.

Reviewer #3: Given that the condition for positive changes resulting from research is to establish a relationship with the community and positive involvement of the community, the present study has positive dimensions in identifying weaknesses related to knowledge, important beliefs of the community and the management of arbovirus diseases in Kenya (perhaps extendable to some other places), and has provided valuable information to address these issues.

PLOS authors have the option to publish the peer review history of their article (what does this mean?). If published, this will include your full peer review and any attached files.

Reviewer #1: No

Reviewer #2: No

Reviewer #3: No

**Key Review Criteria Required for Acceptance?**

**Methods**

-Are the objectives of the study clearly articulated with a clear testable hypothesis stated?

-Is the study design appropriate to address the stated objectives?

-Is the population clearly described and appropriate for the hypothesis being tested?

-Is the sample size sufficient to ensure adequate power to address the hypothesis being tested?

-Were correct statistical analysis used to support conclusions?

-Are there concerns about ethical or regulatory requirements being met?

Reviewer #1: • Sections 2.1 and 2.2: The authors should separate an equation with its supporting sentences from that of the next equation. This will make it easier to follow where one equation ends and the next one starts.

• Page 9: It will be better to replace “sub-counts” with “towns”. If the authors want to retain “sub-counts”, then they should add further details about what they mean.

• On section ‘2.3.Study Area and Sampling Procedure’, the authors should include an Ethics statement. The ethics statement should describe if there was informed consent from the study participants, and how the participants of the study consented to it or how the consent was obtained. It should also explain if there was ethical approval for the study, state the approving organisation(s) and approval number (where applicable).

Reviewer #2: The methodology was very well and in detail described. Potentially it could be made a little more concise as the manuscript overall is quite long, and about 1900 words of 5400 words in total are the methodology. 

Some mentioning of ethical procedure should be added, such as ethical study approval of the institution.

Detailed comments are in the attached document.

**Results**

-Does the analysis presented match the analysis plan?

-Are the results clearly and completely presented?

-Are the figures (Tables, Images) of sufficient quality for clarity?

Reviewer #1: The authors presented the results starting with each Table. The approach is good and easy to follow. However, it seemed as if the table titles were repeated with each table number number. To avoid this repetition and to make the Results sections easier to grasp, it will be better if the authors summarise the main findings of each table at the beginning of the paragraph instead. Then, they can put the Table number being referred to in brackets. Alternatively, the authors can maintain their format but replace the table headings with the summary of the findings of that table.

Reviewer #2: The analysis presented matches the analysis plan.

A very well described and clearly presented result section. 

Tables are very clear.

In Figure 1, do the red spots on the three countys depict the sampled sites or sampled households? As if it were households these seem quite few on the county map, if they were 200 per county? It would additionally make sense to make the county maps bigger, for the sites to be more visible.

**Conclusions**

-Are the conclusions supported by the data presented?

-Are the limitations of analysis clearly described?

-Do the authors discuss how these data can be helpful to advance our understanding of the topic under study?

-Is public health relevance addressed?

Reviewer #1: Discussion

The authors were not generous with the interpretation of their Results. They mostly repeated the Results and sometimes, compared it to previous studies. Few suggestions on how to improve the Discussion section include:

• It will be nicer if the authors started the Discussion with a summary of their results in the study in relation to the aim of their study.

• The Discussion can be broken down into sub-sections that correspond with those in the Results, so that the particular Result being discussed can be easily identified in the article. Alternatively, since the focus of the study is intertwined amongst knowledge, belief and management of arboviral diseases, the authors can separate the Discussion into three groups to tackle the three areas well.

• Summarise a particular finding, discuss what it implies, and then compare if it is consistent or in contrast with the results of other studies. See instances in the comments on page 17.

Conclusion

The authors should indicate the limitation of their study before or after the recommendations.

Reviewer #2: The conclusions is supported by the data presented.

A more extensive section on limitations of analysis should be added to the discussion.

The authors discussed very well the studies public health relevance add how the data can be helpful to advance our understanding of the topic under study.

**Editorial and Data Presentation Modifications?**

Reviewer #1: Summary of the research

The study described the current status of farmers’ (people’s) knowledge, beliefs and management of arboviral diseases. This is mostly a neglected section of disease control, which is very important, especially in decision making. The authors used good techniques for the study and ensured that they wrote for diverse audience, which is commendable. However, the authors need to correct some minor aspects of their manuscript:

• For future purposes, the authors should include line numbers in the manuscript to make it easier to review.

• Kindly alternate the word “majority” with “many” or “most” to avoid redundancy.

• The word “significant” is often linked to the intensity of the results in the Results section. Outside the Results section, it will be better to replace it with similar words like “considerably”, “greatly”.

Authors’ names and affiliations

• The authors should provide full address of their affiliated institutions. For instance, in which town and country is icipe located?

• The email address of the corresponding author shows that he is affiliated to “icipe” but he did not include it as one of his affiliations. Correct accordingly.

Abstract

• The study was on livestock farmers according to the Conclusion section but this was not captured in the aim of the study. So, please rephrase the aim of the study to, “This study evaluated the drivers of livestock farmers’ KBM of ADs from a sample of 629 respondents selected using a three-stage sampling procedure in Kenya’s three hotspot counties of Baringo, Kwale, and Kilifi”.

Keywords

• All the listed keywords are already contained in the title. Kindly replace them with words that are not in the title but are within the abstract to make the manuscript more discoverable. Few suggestions include: public health, livestock farmers, awareness, Rift Valley fever, Chikungunya fever, Dengue fever.

Introduction 

• Page 4: The word, “present” in the last paragraph portrayed the preceding words as humans. Since the authors were trying to describe the content of the remaining sections of the manuscript, “contain” is a better word to use. (Please see the corrections in the MS Word version).

Tables

• Table 3: The authors should insert question mark (?) where necessary under the “characteristics” column.
---

## [Decision Letter · Decision Letter 1]

2 Sep 2021

Dear Mr. Nyangau,

We are pleased to inform you that your manuscript 'Evaluating Livestock Farmers Knowledge, Beliefs, and Management of Arboviral Diseases in Kenya: A Multivariate Fractional Probit Approach' has been provisionally accepted for publication in PLOS Neglected Tropical Diseases.

Best regards,

Alberto Novaes Ramos Jr

Associate Editor

Victor Santana Santos

Deputy Editor

Please consider the minor observations indicated by the reviewers 1 and 2.

Reviewer's Responses to Questions

**Key Review Criteria Required for Acceptance?**

**Methods**

-Are the objectives of the study clearly articulated with a clear testable hypothesis stated?

-Is the study design appropriate to address the stated objectives?

-Is the population clearly described and appropriate for the hypothesis being tested?

-Is the sample size sufficient to ensure adequate power to address the hypothesis being tested?

-Were correct statistical analysis used to support conclusions?

-Are there concerns about ethical or regulatory requirements being met?

Reviewer #1: The authors did a good a good job in their description of the study in such a way that non-experts in the field can understand the message being communicated. There was ethical approval for the study and all participants gave informed consent, which is important for this kind of study.

Reviewer #2: NA

Reviewer #3: (No Response)

**Results**

-Does the analysis presented match the analysis plan?

-Are the results clearly and completely presented?

-Are the figures (Tables, Images) of sufficient quality for clarity?

Reviewer #1: The results were clearly presented.

Reviewer #2: NA

Reviewer #3: (No Response)

**Conclusions**

-Are the conclusions supported by the data presented?

-Are the limitations of analysis clearly described?

-Do the authors discuss how these data can be helpful to advance our understanding of the topic under study?

-Is public health relevance addressed?

Reviewer #1: The conclusion was directly linked to the aim of the study and its implication was elaborated.

Reviewer #2: For line 498 : "However, mosquito nets may offer little protection in reducing the risk of ADs [49,50]. " add a sentence why nets only offer limited protection. Spell out what you are trying to tell the reader, so that reader does not have to guess.

Reviewer #3: (No Response)

**Editorial and Data Presentation Modifications?**

Reviewer #1: On line 37, kindly include "African" to read ...Sub-Saharan African countries.

The authors should consistently use the past tense in the Discussion section. For instance, they should rephrase lines 402-403 to "Individuals who were aware of AD health impacts were more likely to

403 undertake disease mitigation strategies or seek medical intervention".

Reviewer #2: For all citations mentioned in the text, such as in line 469, 504, 564,565 (this is an unexhaustive list) after each citation a dot is needed, when "et al" is mentioned in the text, it should be spelled out as following "et al.".

In line 487, 526 there is double dot after the sentence, which should be reduced to a single dot.

Rephrase Line 493 "young ones among animals" to "young animals".

In line 534: "A household head’s education level increased the intensity of knowledge, beliefs, and management of ADs. ", mention that this is a finding from this study, e.g. "The study identified that a household head’s education level increased the intensity of knowledge, beliefs, and management of ADs.

In line 698 : "Conclusion’s and policy implications" should be rephrased as "Conclusions and policy implications" .

Reviewer #3: (No Response)

**Summary and General Comments**

Reviewer #1: People's knowledge of a disease is very important in the management of that disease. The authors did well to undertake this study and expound on an often understudied area of disease control.

Reviewer #2: The authors addressed the reviewers' comments in a very thorough way and thus have strongly improved the manuscript.

The reviewer had a few very minor points to be addressed by the authors.

Reviewer #3: (No Response)

PLOS authors have the option to publish the peer review history of their article (what does this mean?). If published, this will include your full peer review and any attached files.

Reviewer #1: No

Reviewer #2: No

Reviewer #3: No

---

## [Editor Report · Acceptance letter]

9 Sep 2021

Dear Mr. Nyangau,

We are delighted to inform you that your manuscript, "Evaluating Livestock Farmers Knowledge, Beliefs, and Management of Arboviral Diseases in Kenya: A Multivariate Fractional Probit Approach," has been formally accepted for publication in PLOS Neglected Tropical Diseases.

Best regards,

Shaden Kamhawi

co-Editor-in-Chief

Paul Brindley

co-Editor-in-Chief
